# The dynamics of a duopoly Stackelberg game with marginal costs among heterogeneous players

Atefeh Ahmadi[1], Sourav Roy[2], Mahtab Mehrabbeik[1], Dibakar Ghosh [3]*, Sajad Jafari [1,4], Matjaž Perc [5,6,7,8,9]

1 Department of Biomedical Engineering, Amirkabir University of Technology (Tehran Polytechnic), Tehran, Iran, 2 Department of Mathematics, Jadavpur University, Kolkata, West Bengal, India, 3 Physics and Applied Mathematics Unit, Indian Statistical Institute, Kolkata, West Bengal, India, 4 Health Technology Research Institute, Amirkabir University of Technology (Tehran Polytechnic), Tehran, Iran, 5 Faculty of Natural Sciences and Mathematics, University of Maribor, Maribor, Slovenia, 6 Department of Medical Research, China Medical University Hospital, China Medical University, Taichung, Taiwan, 7 Alma Mater Europaea, Maribor, Slovenia, 8 Complexity Science Hub Vienna, Vienna, Austria, 9 Department of Physics, Kyung Hee University, Seoul, Republic of Korea

* diba.ghosh@gmail.com

**Data Availability Statement:** All relevant data are within the paper.

## Abstract

One of the famous economic models in game theory is the duopoly Stackelberg model, in which a leader and a follower firm manufacture a single product in the market. Their goal is to obtain the maximum profit while competing with each other. The desired dynamics for a firm in a market is the convergence to its Nash equilibrium, but the dynamics of real-world markets are not always steady and can result in unpredictable market changes that exhibit chaotic behaviors. On the other hand, to approach reality more, the two firms in the market can be considered heterogeneous. The leader firm is bounded rationale, and the follower firm is adaptable. Modifying the cost function that affects the firms' profit by adding the marginal cost term is another step toward reality. We propose a Stackelberg model with heterogeneous players and marginal costs, which exhibits chaotic behavior. This model's equilibrium points, including the Nash equilibrium, are calculated by the backward induction method, and their stability analyses are obtained. The influence of changing each model parameter on the consequent dynamics is investigated through one-dimensional and two-dimensional bifurcation diagrams, Lyapunov exponents spectra, and Kaplan-Yorke dimension. Eventually, using a combination of state feedback and parameter adjustment methods, the chaotic solutions of the model are successfully tamed, and the model converges to its Nash equilibrium.

## Introduction

Game situations occur when a series of intelligent decisions are made during their execution, and all these decisions are related to each other and cause effects on each other [1]. As a result, when someone participates in a game, not only do his/her decisions affect the final result, but

**Funding:** M.P. was supported by the Javna Agencija za Raziskovalno Dejavnost RS(Grant Nos. P1-0403 and J1-2457).

also the decisions of other players make a particular impact on the final result of his/her work [2]. This theory focuses on the game, which serves as a model of an interactive situation among rational players [3]. The main point in the game theory is that one player's payoff depends on the strategy implemented by the other players [4–7]. The game defines the identity of the players, the preferences and strategies available, and how these strategies affect the outcome of the players in several social, economic and biological scenarios [8, 9]. According to the game theory, the actions and choices of each participant affect the outcome of the others [10]. Depending on the model in various perspectives, different requirements or assumptions might be required to build up the scenario [11].

Despite many advances, game theory remains to be a nascent and developing science [12]. Game theory is a set of tools that help to make us understand economic factors in strategic situations and also make us see how they work in different situations in an easier way [13]. Game theory is a set of analytical tools economists use to understand strategic situations to increase benefits [14]. A decision-making process is called strategic if one of the players needs to decide based on what other players like, know, believe, and act on a particular note [15]. Strategic behaviors are essential in interactions such as the competition in economic enterprises, multilateral bargaining, and auctions [16, 17]. Nevertheless, since the logic of game theory remains invariant concerning the change in different modes of science, the applications of the game theory can be extended and illustrated metaphorically to portrait situations over several other fields of science [18–25].

Stackelberg competition, or the Stackelberg game, is a strategic game in the economics of sequential and imperfect competition [26]. In economic terms, the imperfection of competition in the market refers to a market in which monopolistic elements and factors are more or less present [27]. In other words, producers and consumers can influence prices in the market to some extent [28]. In this way, the competitors of this game are dual monopoly elements in the market, and the competition is defined within the market [29]. Stackelberg's competition is a strategic game in the economy, where the leading company makes the first move, and following the footsteps of the leading company, follower companies also make their respective moves [30]. This game consists of two players, each with their points [31, 32]. This means that the competition is not about achieving a common goal, and the players compete with each other to improve their situation and increase their profits [33, 34].

Economic markets consist of factories that humans govern [35]. Humans are social creatures and can cooperate under various circumstances [35]. From an evolutionary game standpoint, economic models like the well-known Stackelberg game can be modified to consider these factors [36]. In the classic Stackelberg model, each firm tries to maximize its profit and does not pay any attention to the overall profit of the market, i.e., it tends to make the most possible profit out of the market without considering its consequence on the whole market [26]. However, when cooperation happens, the firms interact in a way that the overall profit of the market becomes the most, even if their profit is not the maximum [36]. This phenomenon can emerge by modifying the cost function of firms.

Since fundamental markets consist of several firms, the Stackelberg model can be extended to networks of interacting firms [37]. Various network structures can affect the profits gained by each company and hence the collective behavior and firms' strategies [38]. Moreover, economic markets are known to be very brutal and do not care about ethics, especially in competitive circumstances [39]. The immoral strategies that firms may choose can be regulated by considering suitable rewards and punishments for firms [39]. These regulatory rules can also be applied in the cost functions. Further justifications can be applied to the cost function to make the goal represented by common profit, not individual profit.

The nonlinear dynamics of the duopoly Stackelberg game with homogeneous bounded rational players were investigated by Peng and Lu [40]. Three years later, this game was upgraded to a game with heterogeneous players, i.e., one bounded rationale and one adaptable, by Xiao et al. [41]. In the following year, Yang et al. [42] introduced the duopoly Stackelberg game with homogeneous players and a modified cost function called the marginal cost. The complex dynamics of the Stackelberg triopoly and duopoly games were also examined by Askar [43, 44]. To the authors' best knowledge, the complete dynamical analysis of a duopoly Stackelberg game with a combination of heterogeneous players and marginal costs has not yet been reported in the literature.

This paper aims to propose a novel game theory model in economics and comprehensively study its complicated behaviors. Our proposed mathematical framework unveils several dynamical aspects, from finding the equilibrium points to their stability analysis. After that, we examine the effects of the incorporated parameters in the model on the state variables by the bifurcation analysis. We also check the effects of the same parameters on the Lyapunov exponent spectra and the Kaplan-Yorke dimension. Following that, we do detailed examinations of the two-dimensional bifurcation diagrams, simultaneously varying two parameters. Also, our work extends up to chaos control by slightly modifying the induced model. The structure of this paper is as follows: The novel duopoly Stackelberg model is introduced in Section 2, and the equilibrium points and their stability analysis, including the Nash equilibrium, are studied. Moreover, the existence of chaos in the proposed model is examined. The parameter variations that influence the model dynamics are investigated in Section 3. Furthermore, the chaos control of the model is implemented successfully in Section 4. At last, the main features of the new model and a summary of the paper are mentioned in Section 5.

## The duopoly Stackelberg model

In order to propose the novel duopoly Stackelberg model, it is essential to introduce each part of it step by step. In the duopoly Stackelberg game, it is assumed that two rival firms manufacture a similar good in a market. One is the leader firm, and the other is the follower. Each firm produces a certain amount of good in a time step ($q_i(t)$) and has a planned amount of production ($Q_i(t)$) as well. It can be concluded that $q_i(t) > 0$, $Q_i(t) > 0$, $i = 1, 2$, $t = 0, 1, 2, \ldots$ Since in reality, there is always a difference between the actual production of a factory and its ideal announced production, this difference can act as a cost function for factories. As the actual production amount approaches the planned amount, the value of the cost function decreases. Many causes influence this inconsistency in the production of factories; however, for simplicity, all of them can be summarized as $c_i$, which is the positive shift parameter. Besides, in this work, a marginal cost ($d_i$) is added to the cost function to mimic the dynamics in economics better. Finally, the cost function is formed as a nonlinear quadratic expression as,

$$C_i(q_i) = c_i(q_i(t) - Q_i(t))^2 + d_i q_i(t), i = 1, 2, t = 0, 1, 2, \ldots \tag{1}$$

Setting $d_i = 0$, we lead to the conventional cost function. The price of a good in a market is affected by the maximum price and the whole amount of a good manufactured by all firms in a market. The total supply $q(t)$ and the price $p(t)$ are calculated through the demand function as,

$$q(t) = q_1(t) + q_2(t), t = 0, 1, 2, \ldots \tag{2}$$

$$p(t) = f(q) = a - bq(t), t = 0, 1, 2, \ldots \tag{3}$$

where, $a$ and $b$ are positive constants. According to Eq (3), the obtained pricing strategy is a

linear function of the total supply. In the competitive market, each firm intends to maximize its profit. A firm's profit is a combination of its income and the cost function. The profit of each firm is expressed by,

$$\pi_i(q_1, q_2) = q_i p - C_i(q_i) = q_i(a - bq) - c_i(q_i - Q_i)^2 - d_i q_i, i = 1, 2 \tag{4}$$

In order to maximize the profit, the firms must detect the amount of production that leads to the most profit. Thus, the derivative of Eq (4), according to $q_i$ is calculated as,

$$\frac{\partial_i(q_1, q_2)}{\partial q_i} = a - b(2q_i + q_j) + 2c_i(Q_i - q_i) - d_i, i = 1, 2, i \neq j \tag{5}$$

In the following step, the resultant equation obtained from Eq (5) should be set to zero, i.e., $\frac{\partial_i(q_1, q_2)}{\partial q_i} = 0$, and also a set of equations should be solved simultaneously, that is,

$$a - b(2q_i + q_j) + 2c_i(Q_i - q_i) - d_i = 0, i = 1, 2, i \neq j \tag{6}$$

The solution of Eq (6) results in a production amount for each of the firms expressed by,

$$\begin{aligned} q_1 &= \frac{a(b + 2c_2) + b(4c_1 Q_1 - 2c_2 Q_2 - 2d_1 + d_2) - 2c_2(d_1 - 2c_1 Q_1)}{3b^2 + 4b(c_1 + c_2) + 4c_1 c_2}, \\ q_2 &= \frac{a(b + 2c_1) + b(4c_2 Q_2 - 2c_1 Q_1 - 2d_2 + d_1) - 2c_1(d_2 - 2c_2 Q_2)}{3b^2 + 4b(c_1 + c_2) + 4c_1 c_2}. \end{aligned} \tag{7}$$

Apart from the model parameters, Eq (7) depends on the planned production amount of each firm. If the optimal amounts of announced production are found, then the optimal amount of Eq (7) can also be obtained. In the duopoly Stackelberg game, the firms plan sequentially but produce goods simultaneously in the market. In other words, the leader firm has to announce its production plan, and then the follower firm tries to determine its production plan according to the leader's plan and make the most profit out of the market. Hence, the leader firm should be smart enough to anticipate the follower's reaction to its announced production plan. In summary, the leader's production plan depends on forecasting the follower's strategy after observing the leader's action. This procedure makes finding the optimal planned production a bit challenging; however, it can be tackled by the backward induction method. In this technique, first, the solutions obtained in Eq (7) are substituted in Eq (4). Then, the optimal $Q_2$ is achieved by setting the derivation of the follower firm profit regarding the planned production to zero, i.e., $\frac{\partial \pi_2(q_1, q_2)}{\partial Q_2} = 0$, which gives, the optimal value of $Q_2$, which is achieved in terms of $Q_1$, presented as,

$$Q_2 = \frac{4(b + c_1)(b + c_2)(a(b + 2c_1)) + b(d_1 - 2(c_1 Q_1 + d_2)) - 2c_1 d_2}{b(8c_2(b + c_1)(b + 2c_1) + b(3b + 4c_1)^2)}. \tag{8}$$

By substituting the optimal value of the follower firm $(Q_2)$, in the profit expression of the leader firm, i.e., $\frac{\partial \pi_1(q_1, q_2)}{\partial Q_1} = 0$, eventually, the optimal value of $Q_1$ is achieved in terms of the model parameters, shown below:

$$Q_1 = \frac{4(b + c_1)(\alpha - bc_2)(a(\alpha - 2bc_1)) + \alpha d_2 - 2d_1(\alpha - bc_2)}{b\beta}, \tag{9}$$

where,

$$\alpha = 3b^2 + 4b(c_1 + c_2) + 4c_1c_2,$$
$$\beta = (9b + 8c_1)\alpha^2 + 8bc_2(b + c_1)(2bc_2 - 3\alpha). \qquad (10)$$

Now, the $Q_1$ obtained in Eq (9) can be replaced in Eq (8), and the final value of $Q_2$ depending just on model parameters is,

$$Q_2 = \frac{(\alpha + b^2)(\alpha(3b(a + d_1 - 2d_2) + 2c_1(a + 2d_1 - 3d_2)) - 4bc_2(b + c_1)(a + d_1 - 2d_2))}{b\beta}. \qquad (11)$$

By substituting Eqs (9) and (11) in the expressions of Eq (7), the optimal production of the two firms is found as,

$$q_1^* = \frac{(\alpha(3b + 4c_1) - 4bc_2(b + c_1))(\alpha(a + d_2 - 2d_1) - 2bc_2(a - d_1))}{b\beta},$$
$$q_2^* = \frac{\alpha(\alpha(3ab + 2a + d_1(3b + 4c_1) - 6d_2(b + c_1)) - 4bc_2(b + c_1)(a + d_1 - 2d_2))}{b\beta}. \qquad (12)$$

where, $(q_1^*, q_2^*)$ are the so-called optimal Nash equilibrium. Finally, after finding all of the terms crucial in establishing the novel model, the duopoly Stackelberg game with marginal costs and heterogeneous players is formed as,

$$q_1(t + 1) = q_1(t)(1 + v_1(a + 2c_1Q_1 - d_1 - 2(b + c_1)q_1(t) - bq_2(t))),$$
$$q_2(t + 1) = q_2(t) + \frac{v_2}{2(b + c_2)}(a + 2c_2Q_2 - d_2 - 2(b + c_2)q_2(t) - bq_1(t)). \qquad (13)$$

In this game, the model is a two-dimensional nonlinear map, and the production of each firm in the next step depends on its current production. It is assumed that the leader firm is bounded rational, i.e., it cannot observe the market interactions entirely but partially. On the other hand, the follower firm has adaptable expectations. As a result, the updating rule of each firm is different from its rival. The updating rule of both firms consists of their current production amount and a portion of their profit variation (the derivation of the profit concerning the production amount calculated in Eq (5). Each firm adapts its production amount to the market change through a constant speed shown by $v_i$ in Eq (13). In the case of zero $v_i$, the firm does not pay any attention to the market trend that may lead to its loss.

### Equilibrium points and stability analysis

Finding equilibrium points in the proposed economic model (13) plays a pivotal role in market analysis. Because if the Nash equilibrium of an economic model is not stable, then the market tends to undesired and uncontrolled variations that remain detrimental to the firms and the customers. Since the proposed model is a map, its equilibrium points can be obtained by setting $q_i(t + 1) = q_i(t)$, $i = 1, 2$. In other words, when a firm's production does not change over time, then this firm is settled in an equilibrium point that can be stable or unstable. Applying this method to the model proposed in Eq (13) results in two distinct solutions. The first production amount that satisfies $q_i(t + 1) = q_i(t)$, $i = 1, 2$, and the obtained equilibrium point is named $E_0 = \left(0, \frac{a + 2c_2Q_2 - d_2}{2(b + c_2)}\right)$. The second equilibrium state is $E_1 = (q_1^*, q_2^*)$, which is

obtained from solving the set of equations,

$$\begin{cases} a + 2c_1Q_1 - d_1 - 2(b + c_1)q_1(t) - bq_2(t) = 0, \\ a + 2c_2Q_2 - d_2 - 2(b + c_2)q_2(t) - bq_1(t) = 0. \end{cases} \quad (14)$$

The set of Eq (14) is similar to Eq (6), and thus, the resultant equilibrium point is the before mentioned Nash equilibrium, $E_1$. The values of the entities of the equilibrium point $E_1$ are identical to Eq (12). Consequently, the proposed model has two different equilibrium points. To determine their stability, the Jacobian matrix of Eq (13) should be formed and is given by,

$$\mathbf{J}(q_1, q_2) = \begin{bmatrix} 1 + v_1(a + 2c_1Q_1 - d_1 - 4(b + c_1)q_1 - bq_2) & -bv_1q_1 \\ \dfrac{-bv_2}{2(b + c_2)} & 1 - v_2 \end{bmatrix}. \quad (15)$$

Then, the characteristic equation of the Jacobian matrix should be shaped. The characteristic equation is written like Eq (16). In Eq (16), $Tr\,\mathbf{J}$ and $Det\,\mathbf{J}$ stand for the trace and the determinant of the Jacobian matrix, respectively. Also, $\lambda$ represents the eigenvalues of the equilibrium point. From the above Jacobian matrix, the characteristic equation be written as,

$$\lambda^2 - Tr(\mathbf{J})\lambda + Det(\mathbf{J}) = 0. \quad (16)$$

where, $Tr(\mathbf{J})$ and $Det(\mathbf{J})$ denote the trace and the determinant of the Jacobian matrix $\mathbf{J}$. The characteristic equation is calculated individually by replacing the state variables, $q_1$ and $q_2$, with the equilibrium points in Jacobian matrix (16). If the absolute value of all eigenvalues is less than 1, i.e., inside the unit circle, the equilibrium point is stable. If one of the eigenvalues is outside the unit circle, then the equilibrium point is unstable. The stability analysis requires in-depth experiments if there is an eigenvalue precisely on the unit circle. First, $(q_1, q_2)$ in the Jacobian matrix are replaced with $E_0$, and the eigenvalues for $E_0$ are given by

$$\begin{aligned} \lambda_1 &= 1 - v_2, \\ \lambda_2 &= 1 + v_1\left[a - d_1 + 2Q_1c_1 - \frac{b(a - d_2 + 2Q_2c_2)}{2(b + c_2)}\right]. \end{aligned} \quad (17)$$

Since $v_i > 0$, $\lambda_1$ is always less than one. Because $q_i(t) > 0$, $i = 1, 2$, $\lambda_2$ is always larger than 1, and hence, $E_0$ is a saddle node and is always unstable. Substituting $E_1$ in Eq (15) results in a Jacobian matrix,

$$\mathbf{J}(E_1) = \begin{bmatrix} 1 - 2v_1(b + c_1)q_1^* & -bv_1q_1^* \\ \dfrac{-bv_2}{2(b + c_2)} & 1 - v_2 \end{bmatrix}. \quad (18)$$

The trace and the determinant of Eq (18) are calculated as,

$$Tr(\mathbf{J}) = 2 - v_2 - 2v_1(b + c_1)q_1^*, \quad (19)$$

$$Det(\mathbf{J}) = 1 - v_2 - v_1\left(2(1 - v_2)(b + c_1) + \frac{b^2v_2}{2(b + c_2)}\right)q_1^*. \quad (20)$$

respectively. If $Tr(\mathbf{J})^2 - 4Det(\mathbf{J})$ of an equilibrium point is equal to or larger than zero, then the stability conditions of that equilibrium point can be determined by the Jury conditions [40].

With the help of Eqs (19) and (20), $Tr(\mathbf{J})^2 - 4Det(\mathbf{J})$ is calculated as,

$$Tr(\mathbf{J})^2 - 4Det(\mathbf{J}) =$$
$$4(v_2 - 1)(b(1 - 2v_1(c_1 + c_2)) + c_2(1 - 2c_1 v_1) - b^2 v_1) + (2b^2 v_1 v_2 + 4(b + c_2)(v_2 - 1))q_1^* \geq 0. \tag{21}$$

By simplifying Eq (21) and noticing that all model parameters are positive, $Tr(\mathbf{J})^2 - 4Det(\mathbf{J})$ leads to a non-negative expression, and hence the Jury conditions can be applied to determine the stability of the Nash equilibrium. Jury conditions specify the stability region of the equilibrium points regarding the model parameters as,

$$\begin{cases} Det(\mathbf{J}) < 1 \\ 1 - Tr(\mathbf{J}) + Det(\mathbf{J}) > 0 \\ 1 + Tr(\mathbf{J}) + Det(\mathbf{J}) > 0 \end{cases} \tag{22}$$

These conditions are shown in Eq (22), and if all of them are satisfied, then the stability of the equilibrium point is assured. In other words, Eq (21) leads to a set of model parameters that makes the equilibrium point of the model stable. These three inequalities bound the parameter space in a certain way that the Nash equilibrium becomes stable. Therefore, they guide appropriately in choosing the model parameters to have a stable Nash equilibrium towards which the firms' dynamics converge. Now, from the results (22),

$$Det(\mathbf{J}) < 1$$

results in inequality,

$$2v_2(b + c_2) + v_1(\alpha(1 - v_2) + b^2)q_1^* > 0, \tag{23}$$

whereas $1 - Tr(\mathbf{J}) + Det(\mathbf{J}) > 0$ produces the inequality,

$$v_1(\alpha + b^2) - (b^2 v_1 + 2(b + c_2))q_1^* > 0. \tag{24}$$

Using $1 + Tr(\mathbf{J}) + Det(\mathbf{J}) > 0$, we reach to the conclusion,

$$(2 - v_2)(4(b + c_2) - v_1(\alpha + b^2)) + (2(b + c_2)(2 - v_2) - b^2 v_1 v_2)q_1^* > 0. \tag{25}$$

## Chaotic behaviors in the model

Fundamental financial markets and businesses are not always stable and have a predictable trend. The unordered behavior of such markets depends on several reasons and complex interactions between the active elements in the market, for instance, factories, consumers, and suppliers. Therefore, the economic model with a stable Nash equilibrium and can also model unpredictable circumstances is very precious. These unpredictable but bounded variations in a market resemble chaotic behavior. To investigate whether Eq (13) can lead to a chaotic solution, the time series of the proposed model and its sensitivity to initial conditions that are symptoms of the existence of chaos are depicted in Fig 1. The x-axis of all panels in Fig 1 is the time step or iteration number, and the y-axis represents the production amount of each firm. The model is simulated for 1000 iterations, and after eliminating the transient parts, the result of the last 100 iterations is plotted. Cyan and yellow dots represent the leader's and the follower's output at each time step. The proposed model can successfully exhibit chaotic behaviors according to the time series pattern. The model parameters are $a = 6$, $b = 0.5$, $c_1 = 2$, $c_2 = 1$, $d_1 = 0.5$, $d_2 = 1$, $v_1 = 0.1$, and $v_2 = 1$. In order to examine the

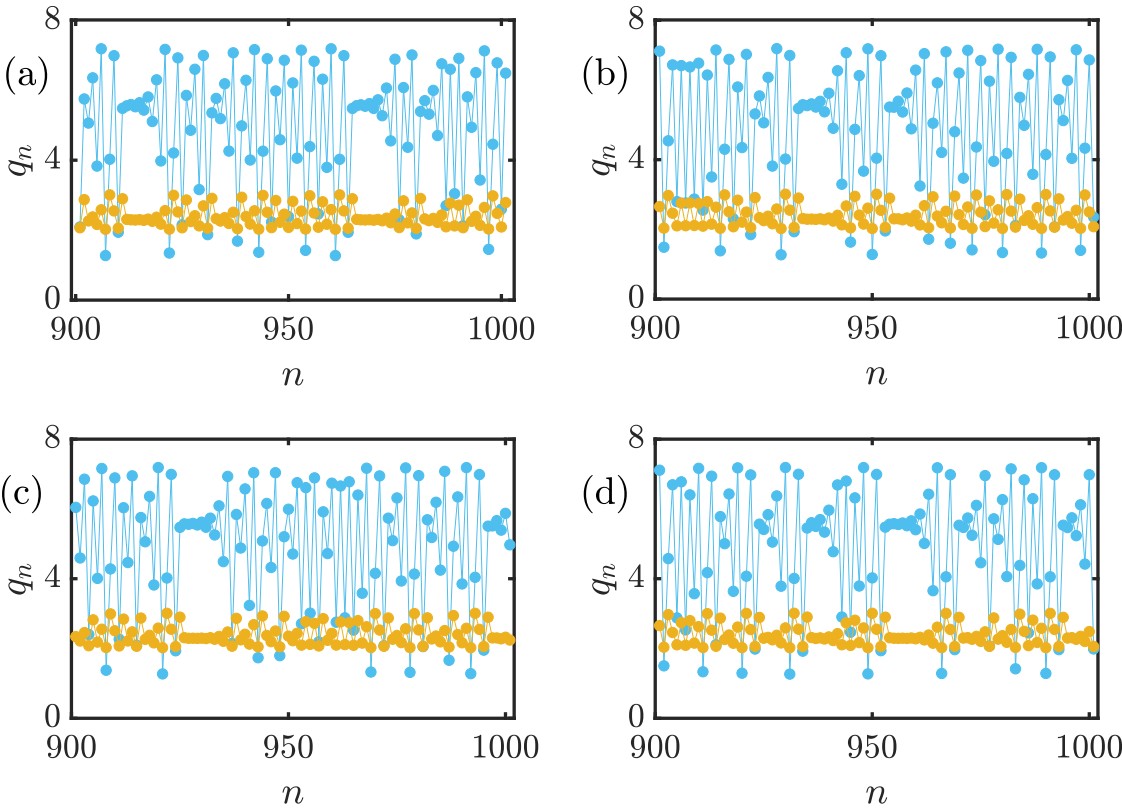

**Fig 1. Time series of Eq (13) and its sensitivity to initial conditions with $a = 6$, $b = 0.5$, $c_1 = 2$, $c_2 = 1$, $d_1 = 0.5$, $d_2 = 1$, $v_1 = 0.1$, and $v_2 = 1$.** The blue and yellow dots represent the output of the leader ($q_1$) and the follower firm ($q_2$). The initial conditions are (a) $q_1(0) = q_2(0) = 1$, (b) $q_1(0) = 1.0001$, $q_2(0) = 1$, (c) $q_1(0) = 1$, $q_2(0) = 1.0001$, and (d) $q_1(0) = 1.0001$, $q_2(0) = 1.0001$. There are bounded but unpredictable and unordered variations in the time series that resemble chaotic behavior. Also, changing the initial conditions in the order of $10^{-4}$ causes an entirely different time series, the sensitive dependence of chaotic models to initial conditions.

sensitive dependence of the model to initial conditions variations, the initial conditions of each panel are set as (a) $q_1(0) = q_2(0) = 1$, (b) $q_1(0) = 1.0001$, $q_2(0) = 1$, (c) $q_1(0) = 1$, $q_2(0) = 1.0001$, and (d) $q_1(0) = 1.0001$, $q_2(0) = 1.0001$. According to Fig 1, with just slightly changing one or both of the initial conditions in the order of $10^{-4}$, the time series of the model completely changes. Nevertheless, these changes occur in a limited region; thus, this bounded time series with sensitivity to initial conditions signify chaotic behavior. The pattern that is shaped by plotting the attractors of the proposed model on the $q_1 - q_2$ plane provides some information about the model's behavior. This model can exhibit chaotic attractors as well as periodic ones. The attractors of the proposed model for different sets of parameters are demonstrated in Fig 2. The constant set of parameters is $c_1 = 2$, $c_2 = 1$, $d_1 = 0.5$, $v_1 = 0.1$, and the initial conditions are $q_1(0) = q_2(0) = 1$. Panel (a) represents a strange attractor with $a = 6$, $b = 0.5$, $d_2 = 1$, and $v_2 = 1.6$. Panel (b) demonstrates another strange attractor with $a = 6$, $b = 0.5$, $d_2 = 0.5$, and $v_2 = 1.6$. Also, panel (c) depicts a new chaotic attractor with $a = 6$, $b = 0.5$, $d_2 = 1$, and $v_2 = 1$. Finally, a period-8 attractor is shown in panel (d) with $a = 6$, $b = 0.6$, $d_2 = 1$, and $v_2 = 1.6$. The topology of the strange attractors changes with the parameters' variations, and each attractor's upper and lower limits are also affected. As a result, the proposed model owns a rich dynamic that motivates further and in detail investigations.

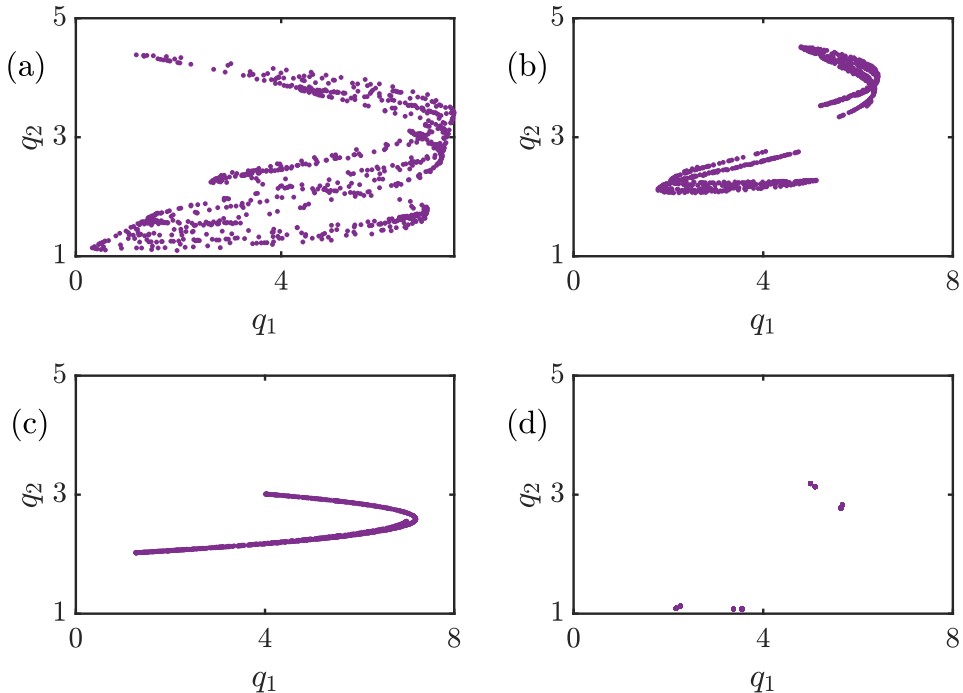

**Fig 2. Periodic and chaotic attractors of Eq (13) on the $q_1 - q_2$ plane with $c_1 = 2$, $c_2 = 1$, $d_1 = 0.5$, $v_1 = 0.1$ and initial conditions $q_1(0) = q_2(0) = 1$.** (a) Strange attractor with $a = 6$, $b = 0.5$, $d_2 = 1$, and $v_2 = 1.6$, (b) Strange attractor with $a = 6$, $b = 0.5$, $d_2 = 0.5$, and $v_2 = 1.6$, (c) Strange attractor with $a = 6$, $b = 0.5$, $d_2 = 1$, and $v_2 = 1$, and (d) Period-8 attractor with $a = 6$, $b = 0.6$, $d_2 = 1$, and $v_2 = 1.6$. This model can not only exhibit various chaotic attractors but also can lead to periodic solutions.

## The effect of parameters on model dynamics

Eq (13) has eight different parameters; some are related to the pricing, and some are associated with the leader and follower firms. Up to now, all parameters have been kept unchanged in the investigations. However, this section aims to study the role of those parameters in the model's dynamics. First, the parameters are considered as the bifurcation parameter one by one, and then the model's dynamics are examined while parameter pairs are changing simultaneously.

### Variation of a single parameter

The influence of every parameter on the model's behavior is studied with the aid of bifurcation diagrams, Lyapunov Exponents spectra, and the Kaplan-Yorke dimension. Consider calculating a system's Lyapunov exponents; if the Lyapunov exponents are sorted descending, the value of the corresponding Kaplan-Yorke dimension is achieved as, [45, 46].

$$D_{KY} = j + \frac{\sum_{i=1}^{j} \lambda_i}{|\lambda_{j+1}|},$$

$$\sum_{i=1}^{j} \lambda_i \geq 0, \tag{26}$$

$$\sum_{i=1}^{j+1} \lambda_i < 0.$$

In this equation, $\lambda_i$ represents the Lyapunov exponents, and $j$ is the index of that Lyapunov

exponent up to which the summation of all Lyapunov exponents is non-negative, but after that, the summation becomes negative. The Kaplan-Yorke dimension quantifies the dimension of the space in which a system's attractor exists. In other words, it indicates the complexity of attractors. Eq (13) is a two-dimensional map, so it results in two Lyapunov exponents that are assumed as $\lambda_1 \geq \lambda_2$. While the attractors of the model are fixed points or periodic, both of the Lyapunov exponents are negative, which leads to a zero Kaplan-Yorke dimension. However, since a positive Lyapunov exponent exists in the chaotic region, the Kaplan-Yorke dimension can vary between one and two. Due to the negativity of the other Lyapunov exponent that always has a more significant absolute value, the Kaplan-Yorke dimension of a strange attractor cannot be equal to or larger than 2.

The following figures in this section, panels (a)-(c), represent the bifurcation diagram, Lyapunov exponents spectra, and the Kaplan-Yorke dimension. The red line in panel (b) shows the largest, and the green line shows the other Lyapunov exponent. All numerical simulations are performed with fixed initial conditions as $q_1(0) = q_2(0) = 1$. First, the model's behavior under the variation of $a$ is investigated. By varying $a$ in the $[0, 6.4]$ interval, its influence on the model's dynamic with $b = 0.5$, $c_1 = 2$, $c_2 = 1$, $d_1 = 0.5$, $d_2 = 1$, and $v_1 = 0.1$, and $v_2 = 1$ is shown in Fig 3. Here, $a$ is the maximum price in the market that affects the demand function. According to Fig 3, it seems that by growing the maximum price of a good in the market, the market behavior tends to be chaotic that is desired neither by the companies nor the customers. Also, this chaoticity in firms' production amount leads to the unpredictability of the market trend. Consequently, the firms cannot plan appropriately, so the possibility of a crisis increases. Some

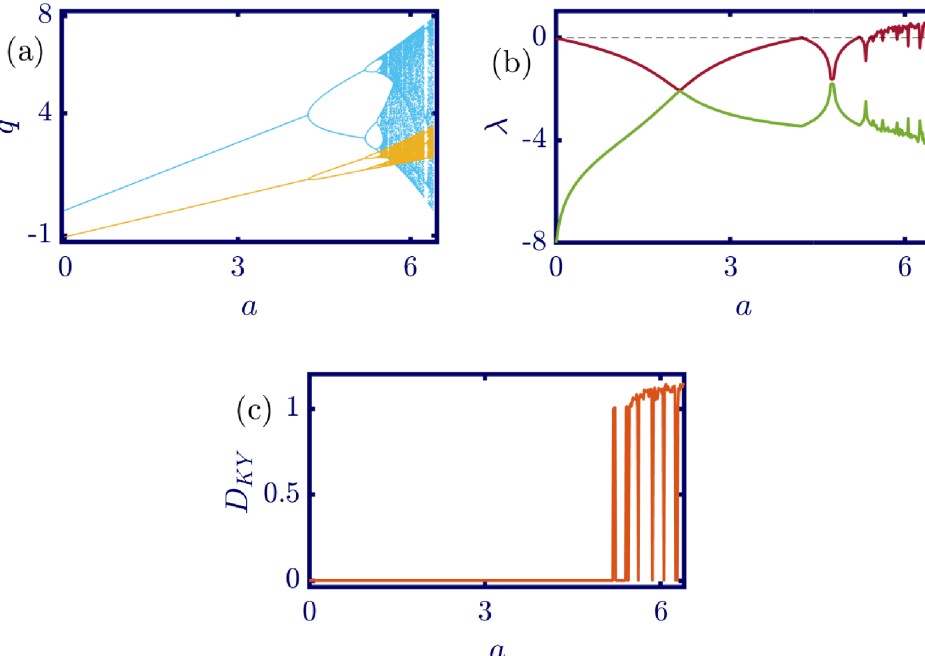

**Fig 3. Eq (13) dynamics concerning parameter $a$ with $b = 0.5$, $c_1 = 2$, $c_2 = 1$, $d_1 = 0.5$, $d_2 = 1$, $v_1 = 0.1$, $v_2 = 1$ and initial conditions $q_1(0) = q_2(0) = 1$.** (a) Bifurcation diagram, (b) Lyapunov exponents spectra, and (c) Kaplan-Yorke dimension. With the increase in the value of the parameter $a$ which is the maximum price of a good in the demand function, the market exhibits chaotic behavior that, due to its unpredictability, is desired neither by the firms nor the consumers. In the sub-figure (a), the cyan points represent the data for the leader firm ($q_1$), whereas the yellow points represent those for the follower firm ($q_2$), and in sub-figure (b), the red and green points present the Lyapunov exponent spectra for the leader and follower firms respectively, namely $\lambda_1$ and $\lambda_2$.

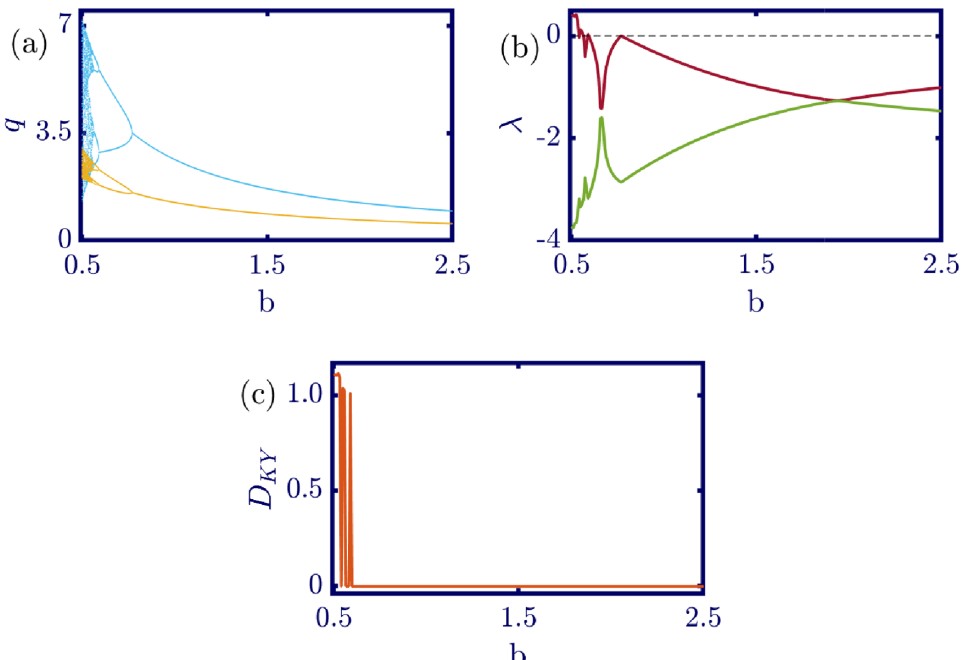

**Fig 4. Eq (13) dynamics concerning parameter $b$ with $a = 6$, $c_1 = 2$, $c_2 = 1$, $d_1 = 0.5$, $d_2 = 1$, $v_1 = 0.1$, $v_2 = 1$ and initial conditions $q_1(0) = q_2(0) = 1$.** (a) Bifurcation diagram, (b) Lyapunov exponents spectra, and (c) Kaplan-Yorke dimension. The chaotic behavior is only observable for a small range of the parameter. Thus, the stability of the market can be easily achieved by selecting parameter $b$ in a wide range of values that results in the Nash equilibrium. In the sub-figure (a), the cyan points represent the data for the leader firm ($q_1$), whereas, the yellow points represent those for the follower firm ($q_2$), and in sub-figure (b), the red and green points present the Lyapunov exponent spectra for the leader and follower firms respectively, namely $\lambda_1$ and $\lambda_2$.

thin periodic windows are seen in the middle of the chaotic region but are very narrow and temporary. In conclusion, the period doubling route to chaos and positive Lyapunov exponent in the chaotic regime is observed by the variation of parameter $a$. The demand function has two parameters: $a$ and $b$. Parameter $a$ has been studied, and now parameter $b$ is considered. The results of changing parameter $b$ in the [0.5, 4.6] interval with $a = 6$, $c_1 = 2$, $c_2 = 1$, $d_1 = 0.5$, $d_2 = 1$, $v_1 = 0.1$, and $v_2 = 1$ are depicted in Fig 4. It can be perceived that only in a small interval of parameter $b$ the chaotic behavior is possible. As a result, by appropriately selecting the value of parameter $b$ in the demand function, the unpredictable behavior of the market is avoidable, and the market experiences stability. Besides, by increasing the value of parameter $b$, the inverse period-doubling bifurcation is seen in panel (a) of Fig 4. The effect of the cost coefficient of the leader firm ($c_1$) is examined in Fig 5. Considering $c_1$ as the bifurcation parameter and $a = 6$, $b = 0.5$, $c_2 = 1$, $d_1 = 0.5$, $d_2 = 1$, $v_1 = 0.1$, and $v_2 = 1$, the period doubling route to chaos is observed in panel (a) of Fig 5. In other words, by increasing the value of $c_1$ the importance of the difference between the announced production amount and the current production amount becomes crucial. This difference is due to not appropriately scheduling the production amount and failing to predict the rival company's plan. Highlighting this difference raises the possibility of falling into the chaotic region. Hence, precise observation of the market trend and the decisions of the rival firm is a critical point in economic management. Since there is another competing company in the market called the follower firm, its cost coefficient's influence on the market dynamics should be noticed. The results of changing $c_2$ in the interval of [0, 10] with $a = 6$, $b = 0.5$, $c_1 = 2$, $d_1 = 0.5$, $d_2 = 1$, $v_1 = 0.1$, and $v_2 = 1$ is demonstrated in Fig 6.

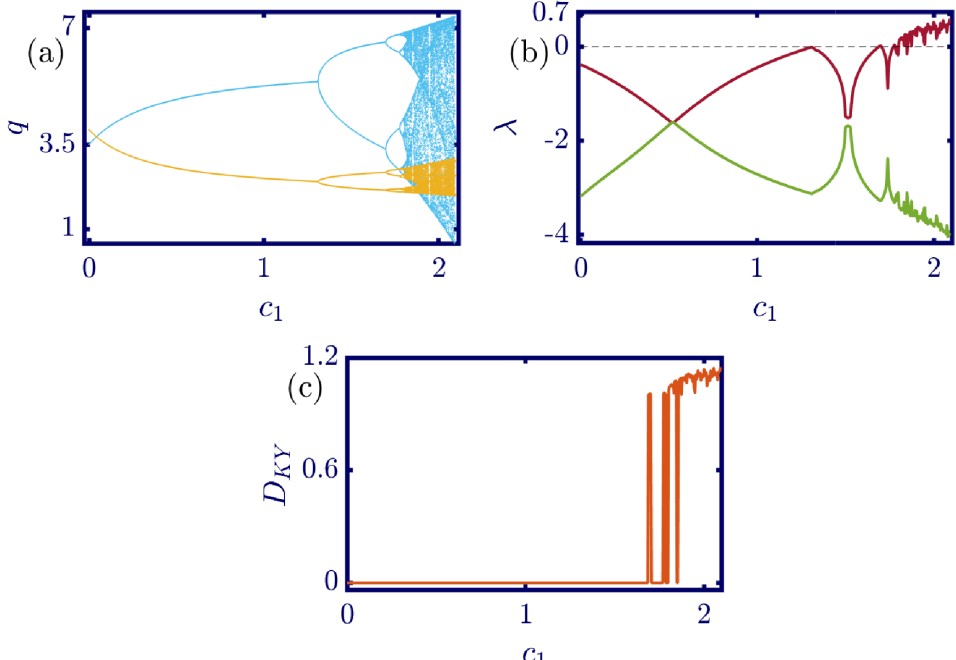

**Fig 5. Eq (13) dynamics concerning the parameter $c_1$ with $a = 6$, $b = 0.5$, $c_2 = 1$, $d_1 = 0.5$, $d_2 = 1$, $v_1 = 0.1$, $v_2 = 1$ and initial conditions $q_1(0) = q_2(0) = 1$.** (a) Bifurcation diagram, (b) Lyapunov exponents spectra, and (c) Kaplan-Yorke dimension. By increasing the cost coefficient, the difference between the planned and current amount of production becomes more significant. This un-organization in planning the company's manufacturing strategy results in a chaotic and uncontrollable market. In the sub-figure (a), the cyan points represent the data for the leader firm ($q_1$), whereas, the yellow points represent those for the follower firm ($q_2$), and in sub-figure (b), the red and green points present the Lyapunov exponent spectra for the leader and follower firms respectively, namely $\lambda_1$ and $\lambda_2$.

Surprisingly, for almost all of the parameter values, both firms are chaotically acting with tiny periodic windows. Nevertheless, there is a notable difference between the leader's behavior and the follower firm; the leader firm's oscillation ranges are roughly constant, but the follower firm is falling. For these specific sets of parameters, the dynamic of both companies is usually chaotic, with some exceptions in the periodic windows. The cost function used in this work is a modified version of the conventional quadratic cost function. this modification is done by adding a term to the definition of the cost function, which is named marginal cost. By changing the marginal cost of the leader firm ($d_1$) in the [0.3, 3.4] interval with $a = 6$, $b = 0.5$, $c_1 = 2$, $c_2 = 1$, $d_2 = 1$, $v_1 = 0.1$, and $v_2 = 1$, the dynamics of the model are portrayed in Fig 7. Increasing this marginal cost results in a period halving route to chaos. It can be inferred that when the value of the marginal cost is not large enough, the cost function is not shaped suitably, so the market becomes unstable for the firms, leading them to chaotic behavior. In summary, the marginal cost should be large enough to make the cost function work properly and avoid unforeseen oscillations. The general cost function for both firms contains marginal cost, so the way the model's dynamic is affected by the marginal cost of the follower firm, namely $d_2$, should be studied. While varying $d_2$ in the interval of [0, 1.4] with $a = 6$, $b = 0.5$, $c_1 = 2$, $c_2 = 1$, $d_1 = 0.5$, $v_1 = 0.1$, and $v_2 = 1$ the resultant dynamics are illustrated in Fig 8. In contrast with Fig 7, the bifurcation diagram in Fig 8 suggests a period-doubling route to chaos, which means that for the follower firm, smaller values of the marginal cost play a better role in converging to the Nash equilibrium. Furthermore, the parameter interval consists of a vast chaotic region with broader periodic windows that make wisely selecting the marginal cost of the follower

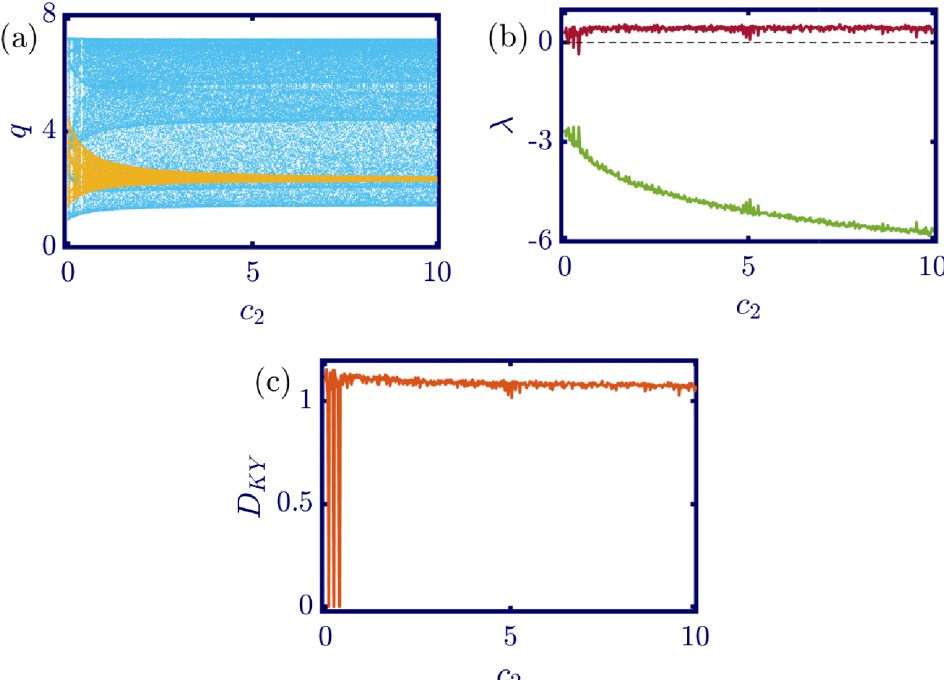

**Fig 6. Eq (13) dynamics concerning the parameter $c_2$ with $a = 6$, $b = 0.5$, $c_1 = 2$, $d_1 = 0.5$, $d_2 = 1$, $v_1 = 0.1$, $v_2 = 1$ and initial conditions $q_1(0) = q_2(0) = 1$.** (a) Bifurcation diagram, (b) Lyapunov exponents spectra, and (c) Kaplan-Yorke dimension. In almost all parameter intervals except some narrow periodic windows, the behavior of both competing firms is chaotic, with a difference in the amplitude of oscillations. The follower firm oscillates in a smaller range, and its amplitude decreases with the bifurcation parameter. In the sub-figure (a), the cyan points represent the data for the leader firm ($q_1$), whereas, the yellow points represent those for the follower firm ($q_2$), and in sub-figure (b), the red and green points present the Lyapunov exponent spectra for the leader and follower firms respectively, namely $\lambda_1$ and $\lambda_2$.

firm essential. The last parameter that affects the market dynamics and should be considered is the speed of the firms in adapting to the market variations. When the parameters are set as $a = 6$, $b = 0.5$, $c_1 = 2$, $c_2 = 1$, $d_1 = 0.5$, $d_2 = 1$, and $v_2 = 1$, and the adaptation speed of the leader firm acts as the bifurcation parameter; the consequent diagrams are shown in Fig 9. The period doubling route to chaos leads the firms to an undesired market. At first glance, it may become apparent that a more considerable adaptation speed is better for a company because it shows flexibility. However, it should be noted that this high flexibility is sometimes not only the strength of a company but also its weakness. Since the leader firm's adaptation speed is under investigation, this company must have the power to affect and control the market, not be defeated by the market. Thus, according to the results depicted in Fig 9, the leader firm should have tiny values of adaptation speed with the market not to be influenced by the improper dynamics of the market. In the way of studying the role of adaptation speed in the model's dynamics, the dynamical analysis is conducted regarding $v_2$ with $a = 6$, $b = 0.5$, $c_1 = 2$, $c_2 = 1$, $d_1 = 0.5$, $d_2 = 1$, and $v_1 = 0.1$ in Fig 10. It can be deduced that the firms are manufacturing products in a chaotic market for almost all adaptation speed values except the periodic windows. Since the follower firm is not the dominant firm in the market, changing its adaptation speed does not influence the market considerably. In other words, the growth in its adaptation speed causes its more conformance to the market variations that are not always appropriate trends. In summary, in comparison with the leader firm, the follower firm plays the role of an enslaved factory in the market. This inference can be recognized from the increasing

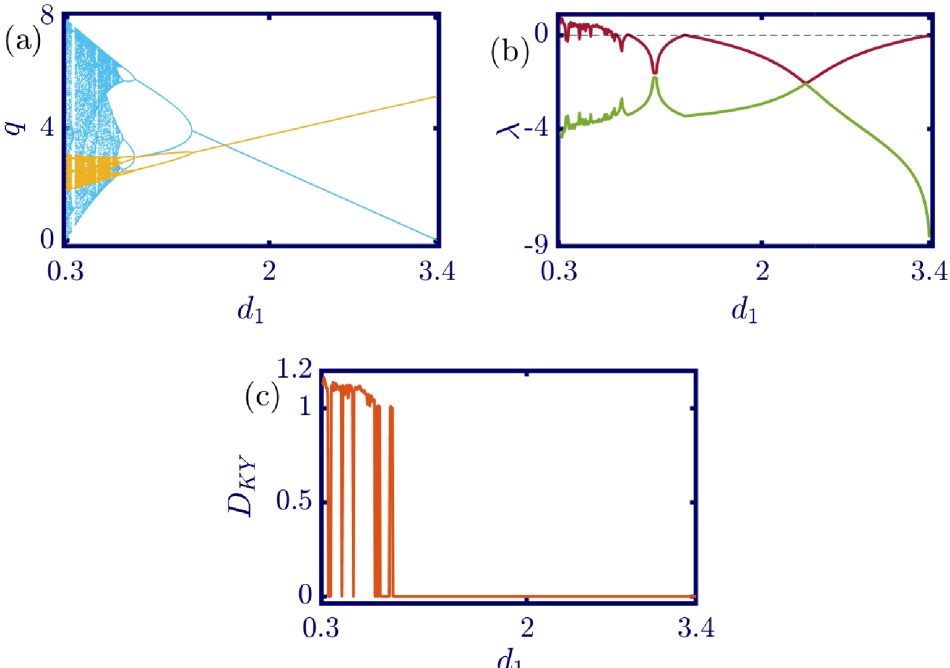

**Fig 7. Eq (13) dynamics concerning the parameter $d_1$ with $a = 6$, $b = 0.5$, $c_1 = 2$, $c_2 = 1$, $d_2 = 1$, $v_1 = 0.1$, $v_2 = 1$ and initial conditions $q_1(0) = q_2(0) = 1$.** (a) Bifurcation diagram, (b) Lyapunov exponents spectra, and (c) Kaplan-Yorke dimension. A periodic halving route to chaos is observed by changing the marginal cost of the leader firm. As a result, avoiding a chaotic and unstable market requires selecting a sufficiently large value for marginal cost. This marginal cost acts like a punishment in the company's cost function and forces it to converge to the Nash equilibrium. In the sub-figure (a), the cyan points represent the data for the leader firm ($q_1$), whereas the yellow points represent those for the follower firm ($q_2$), and in sub-figure (b), the red and green points present the Lyapunov exponent spectra for the leader and follower firms respectively, namely $\lambda_1$ and $\lambda_2$.

amplitude of the follower firm's oscillations in the bifurcation diagram, although the amplitude of the leader firm's oscillations is approximately constant.

## Variation of two parameters simultaneously

The evolution of the model's dynamics regarding all of the model's parameters has been analyzed in detail. In this subsection, the various behaviors of Eq (13) while changing pairs of parameters are studied through two-dimensional bifurcation diagrams. Such color-coded bifurcation diagrams are illustrated in Fig 11. All panels are obtained using $q_1(0) = q_2(0) = 1$ as initial conditions and a $1000 \times 1000$ grid of bifurcation parameters. In each panel, two of the model's parameters are variable, and the rest are constants with values equal to $a = 6$, $b = 0.5$, $c_1 = 2$, $c_2 = 1$, $d_1 = 0.5$, $d_2 = 1$, $v_1 = 0.1$, and $v_2 = 1$. Panels (a)-(d) portray the two-dimensional bifurcation diagrams in the $a - b$, $c_1 - c_2$, $d_1 - d_2$, and $v_1 - v_2$ parameter spaces, respectively. Various solutions of the model are coded by a color map in which the dark blue region is associated with the fixed point solution, and the yellow zone represents chaotic behavior. Different periodic attractors belong to colors in the middle of the color spectrum. As the period of the oscillations increases, the color related to it transforms from blue to green and eventually yellow. The white regions show the unbounded solutions of the model. In all of the panels of Fig 11, both fixed points and strange attractors are observable, which means that the proposed model can successfully imitate the natural dynamics of the economic markets. Moreover, the dominant behavior in all of the panels of Fig 11 is the convergence to a stable fixed point, that

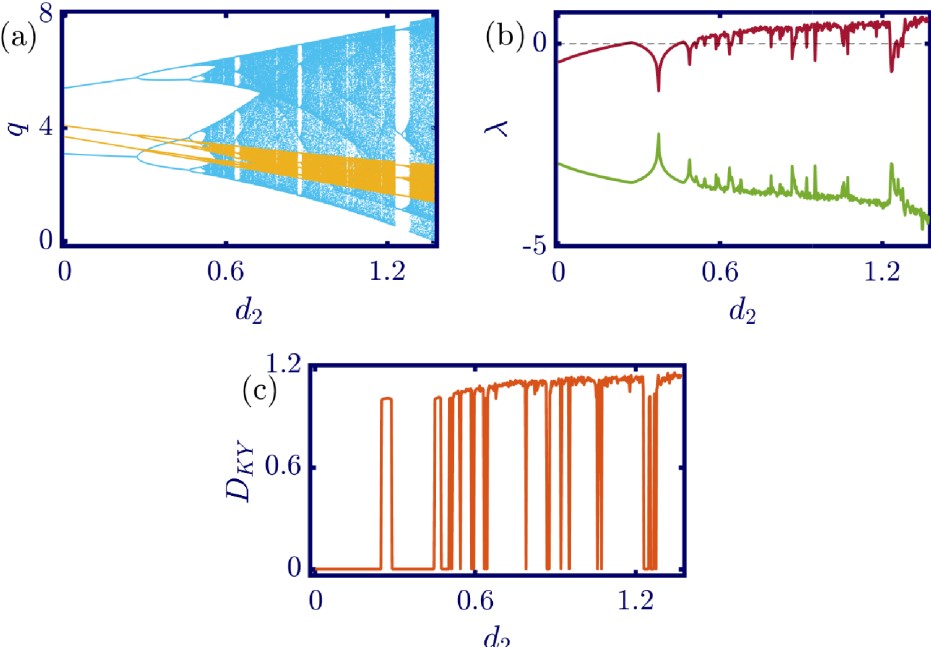

**Fig 8. Eq (13) dynamics concerning the parameter $d_2$ with $a = 6$, $b = 0.5$, $c_1 = 2$, $c_2 = 1$, $d_1 = 0.5$, $v_1 = 0.1$, $v_2 = 1$ and initial conditions $q_1(0) = q_2(0) = 1$.** (a) Bifurcation diagram, (b) Lyapunov exponents spectra, and (c) Kaplan-Yorke dimension. In contrast with the leader firm, due to the period halving route to chaos in the bifurcation diagram, minor marginal cost values better affect the outcome of the follower firm. Since only a limited range of the marginal costs leads to the Nash equilibrium, this parameter should be chosen wisely for the follower firm. In the sub-figure (a), the cyan points represent the data for the leader firm ($q_1$), whereas, the yellow points represent those for the follower firm ($q_2$), and in sub-figure (b), the red and green points present the Lyapunov exponent spectra for the leader and follower firms respectively, namely $\lambda_1$ and $\lambda_2$.

is, the Nash equilibrium of the model. This response is the desired response of the proposed model because, in the Nash equilibrium, all players are in their best situation, considering the game as a whole. For some specific values of parameters, the proposed model exhibits chaotic solutions that are undesired but possible and sometimes unavoidable in economics.

## Chaos control

Due to the complex dynamics and inherent instability of the chaotic models, controlling them so that they exhibit the desired behavior seems impossible. However, it has been shown in various references that chaotic duopoly models are controllable, and different control goals can be imagined for them [47–49]. Since it has been shown that the proposed model can put forward complex, chaotic behaviors, and this model is a sample of a market, unpredictable behaviors are unsuitable for the firms working in this market. The desired dynamic for economic markets is convergence to their Nash equilibrium. This part aims to apply a chaos control method to the proposed model to remove it from the chaotic region. The control method used for the current work is a combination of state feedback and parameter adjustment techniques [50]. In order to implement the mentioned control method, we propose model,

$$\begin{cases} q_1(t+1) = (1-\gamma)q_1(t)(1 + v_1(a + 2c_1Q_1 - d_1 - 2(b+c_1)q_1(t) - bq_2(t))) + \gamma q_1(t), \\ q_2(t+1) = (1-\gamma)q_2(t) + \dfrac{v_2(1-\gamma)}{2(b+c_2)}(a + 2c_2Q_2 - d_2 - 2(b+c_2)q_2(t) - bq_1(t)) + \gamma q_2(t). \end{cases} \quad (27)$$

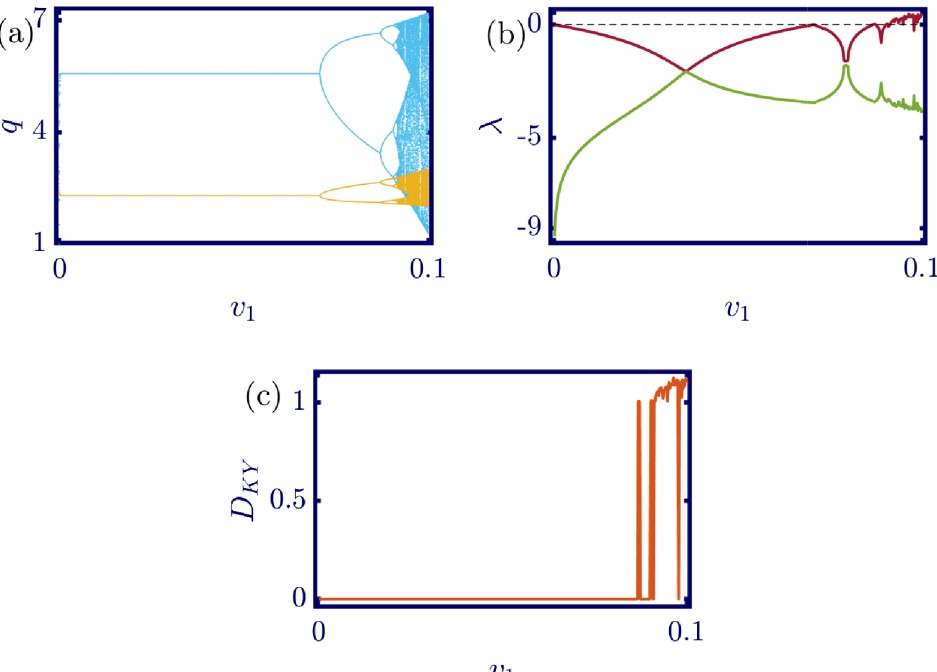

**Fig 9. Eq (13) dynamics concerning the parameter $v_1$ with $a = 6$, $b = 0.5$, $c_1 = 2$, $c_2 = 1$, $d_1 = 0.5$, $d_2 = 1$, $v_2 = 1$ and initial conditions $q_1(0) = q_2(0) = 1$.** (a) Bifurcation diagram, (b) Lyapunov exponents spectra, and (c) Kaplan-Yorke dimension. According to the period doubling route to chaos, smaller values of adaptation speed are desired for the leader firm. The leader firm should control the market, not vice versa. If the opposite occurs, it may lead to the unpredictability of the market. In the sub-figure (a), the cyan points represent the data for the leader firm ($q_1$), whereas, the yellow points represent those for the follower firm ($q_2$), and in sub-figure (b), the red and green points present the Lyapunov exponent spectra for the leader and follower firms respectively, namely $\lambda_1$ and $\lambda_2$.

In the above model, $\gamma$ is the control parameter that can vary between zero and one. Setting $\gamma = 0$ makes Eq (27) identical to Eq (13), which is the original and uncontrolled model. $\gamma = 1$ leads to $q_i(t + 1) = q_i(t)$, $i = 1, 2$ forcefully that means the model must converge to its stable equilibrium point (if any).

Taking $\gamma$ as the bifurcation parameter, the effect of varying its value on the model dynamics is studied in Fig 12. All other model parameters are constant as $a = 6$, $b = 0.5$, $c_1 = 2$, $c_2 = 1$, $d_1 = 0.5$, $d_2 = 1$, $v_1 = 0.1$, $v_2 = 1$, and the initial conditions are $q_1(0) = q_2(0) = 1$. By increasing the value of $\gamma$ from zero to one, the period halving bifurcation occurs. In other words, with weak value of $\gamma$, the proposed model is chaotic and still uncontrolled, but gradually the chaotic behavior transforms into periodic solutions, and finally, at the critical point $\gamma_c = 0.3$, the model converges to its calculated Nash equilibrium, which is stable with this set of parameters. Therefore, the sufficient amount of control parameter $\gamma$ leads to the negativity of the largest Lyapunov exponent of the model and the zero dimension of its attractor that has been changed to a fixed point. So, the proposed model can be controlled successfully with the referenced procedure.

## Discussion and conclusion

One of the complex economic game models introduced in this article is the duopoly Stackelberg game model. This model consists of two competing firms; one acts as the leader, and the other is the follower. These firms manufacture similar products in a common market and compete to maximize profits. If an economic model can imitate the actual dynamics of a

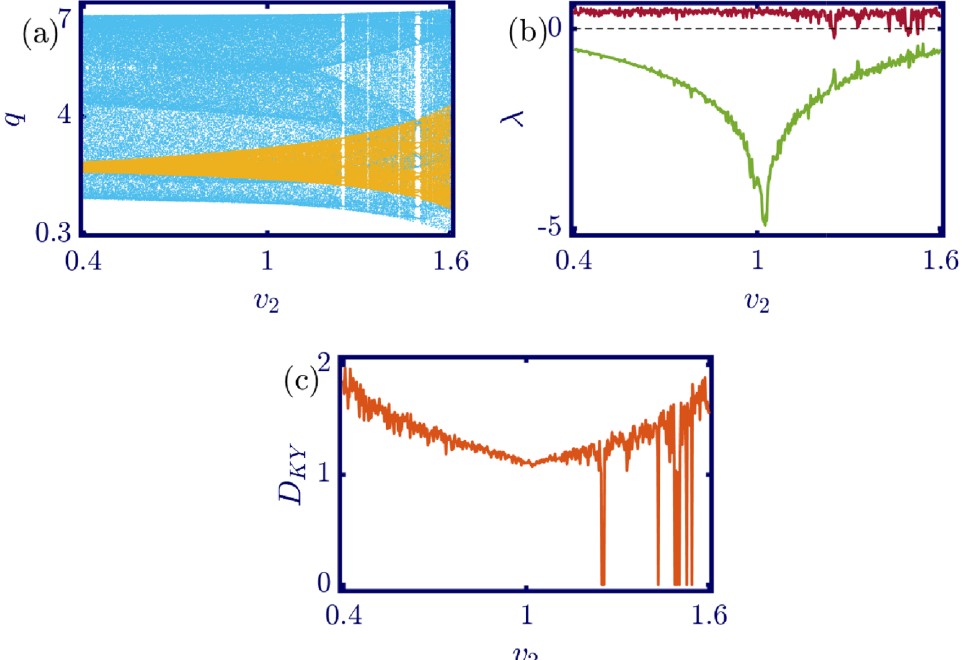

**Fig 10. Eq (13) dynamics concerning the parameter $v_2$ with $a = 6$, $b = 0.5$, $c_1 = 2$, $c_2 = 1$, $d_1 = 0.5$, $d_2 = 1$, $v_1 = 0.1$ and initial conditions $q_1(0) = q_2(0) = 1$.** (a) Bifurcation diagram, (b) Lyapunov exponents spectra, and (c) Kaplan-Yorke dimension. Even though some periodic windows exist, chaotic behavior is the dominant dynamic in the whole parameter interval. It can be deduced that increasing the adaptation speed of the follower firm not only does not control the market but also raises its flexibility and conformity with the market, which are not always appropriate behaviors. In the sub-figure (a), the cyan points represent the data for the leader firm ($q_1$), whereas, the yellow points represent those for the follower firm ($q_2$), and in sub-figure (b), the red and green points present the Lyapunov exponent spectra for the leader and follower firms respectively, namely $\lambda_1$ and $\lambda_2$.

market, then that model is valuable. In order to approach the fundamental behaviors observed in economic markets, a duopoly Stackelberg model with heterogeneous players and marginal costs was introduced in this paper that has not yet been reported in other related research. Marginal costs are a modification of the cost function of the firms, and heterogeneous players mean that the leader firm is bounded rational, although the follower firm is adaptable. The new model was constructed, and its two equilibrium points, including the Nash equilibrium, were calculated. According to the possible values of the model's parameters and considering some facts about the nature of this game, it was found that one of the equilibrium points is always unstable, and the Nash equilibrium can be stable under the Jury conditions.

Economic markets can experience desired and undesired dynamics. Convergence to the Nash equilibrium is required, whereas obtaining chaos in the system is unwelcome. Despite this preference, fundamental economic markets experience both of the mentioned behaviors. Therefore, a practical model should be able to model various dynamics. The existence of chaos in the proposed model was approved by plotting the model's time series, and strange hidden attractors [51]. Besides, the sensitive dependence of the model on initial conditions was confirmed. The proposed model has eight parameters, is very flexible, and has considerable freedom. The influence of varying each parameter individually on the dynamics of the model was investigated through bifurcation diagrams, Lyapunov exponents spectra, and the Kaplan-Yorke dimension. Furthermore, the two-dimensional bifurcation diagrams of the proposed model while changing the value of different pairs of parameters were examined. These powerful tools revealed some critical points about the nature of the economic markets that are not

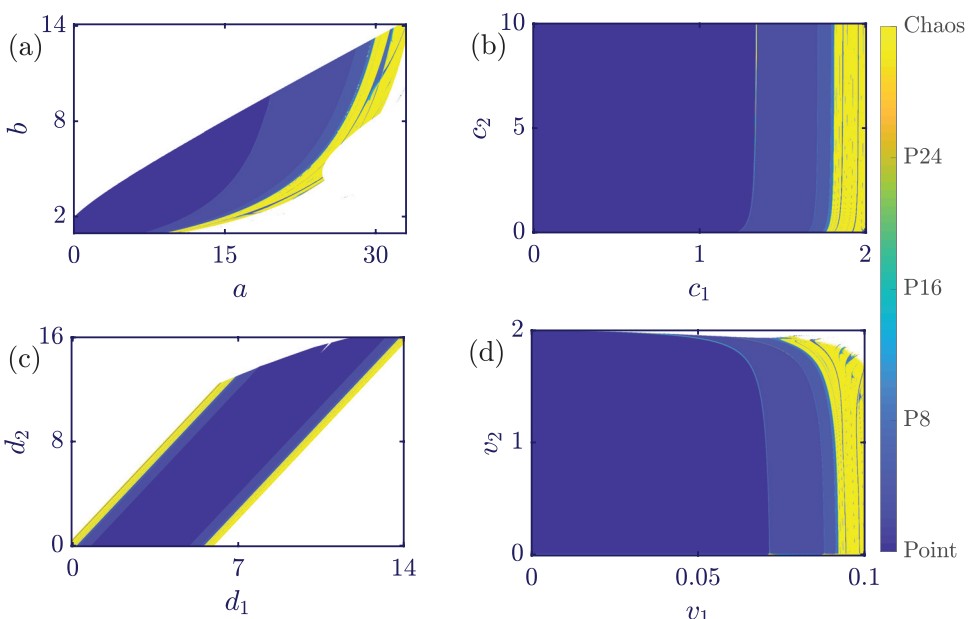

**Fig 11. Two-dimensional bifurcation diagrams of Eq (13) with initial conditions $q_1(0) = q_2(0) = 1$ in (a) $a - b$, (b) $c_1 - c_2$, (c) $d_1 - d_2$, and (d) $v_1 - v_2$ parameter spaces.** The white, yellow, and dark blue regions represent unbounded, chaotic, and fixed point solutions. Each panel is obtained by simulating a grid of $1000 \times 1000$ parameters that are changing, and the remaining parameters are kept fixed at $a = 6$, $b = 0.5$, $c_1 = 2$, $c_2 = 1$, $d_1 = 0.5$, $d_2 = 1$, $v_1 = 0.1$, and $v_2 = 1$. By increasing the number of periods in a periodic attractor, the blue component of its representative color decreases, and the yellow component dominates. Both convergences to the Nash equilibrium and chaotic behaviors can be perceived in all of the panels, which shows the superior capability of the proposed model.

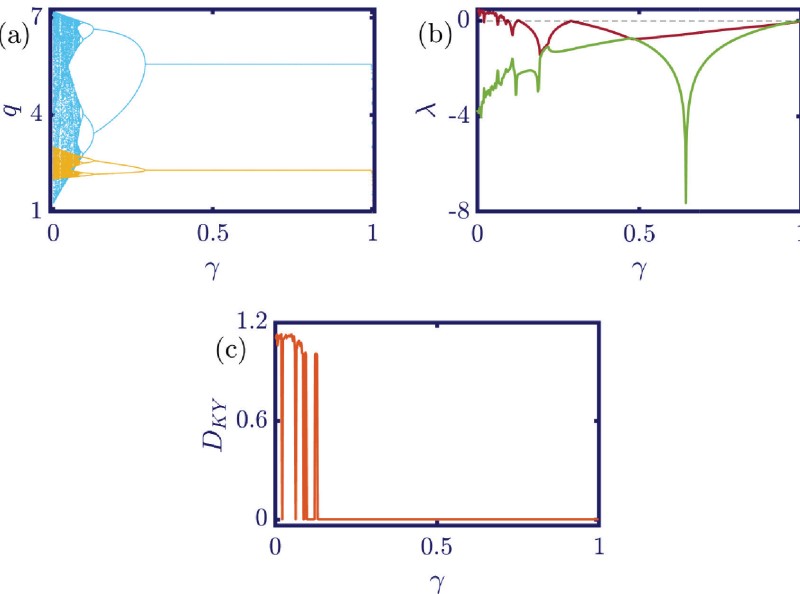

**Fig 12. Eq (27) dynamics concerning the control parameter $\gamma$ with $a = 6$, $b = 0.5$, $c_1 = 2$, $c_2 = 1$, $d_1 = 0.5$, $d_2 = 1$, $v_1 = 0.1$, $v_2 = 1$ and initial conditions $q_1(0) = q_2(0) = 1$.** (a) Bifurcation diagram, (b) Lyapunov exponents spectra, and (c) Kaplan-Yorke dimension. With the increase in the value of $\gamma$ the behavior of the controlled system changes from chaotic to periodic gradually and finally to a fixed point. The inverse period doubling is seen and after the critical point $\gamma_c = 0.3$ the control parameter is large enough to force the model to converge to its stable Nash equilibrium. In the sub-figure (a), the cyan points represent the data for the leader firm ($q_1$), whereas, the yellow points represent those for the follower firm ($q_2$), and in sub-figure (b), the red and green points present the Lyapunov exponent spectra for the leader and follower firms respectively, namely $\lambda_1$ and $\lambda_2$.

trivial at first glance. Moreover, due to the undesirability of chaotic behaviors in the market, a chaos control technique based on state feedback and parameter adjustment has been implemented in the proposed model that can successfully rescue the model from chaotic oscillations.

In general, the Stackelberg model's underlying properties have not yet been fully discovered. Most of the research focus on the duopoly game and some rare works have been done on the triopoly game. Nevertheless, the actual markets consist of more than two or three competing companies. Therefore, one future research line can be extending the sparse markets by considering more firms. However, as the number of firms increase, the backward induction technique becomes time-consuming and so complicated that it cannot be solved analytically. After tackling this obstacle, this economic model can even be generalized to complex networks where collective behaviors can be investigated. As the first step, the heterogeneous players and marginal costs can be applied to the triopoly case and more realistic markets. Besides, from the control theory viewpoint, the performance of various control techniques can be examined to determine the best procedure for guiding the market to the desired state. In conclusion, vast potential future works can be conducted based on this economic game.

## Supporting information

**S1 File.**
(STY)

**S2 File.**
(BST)

**S3 File.**
(CLO)

**S4 File.**
(CLS)

**S5 File.**
(CLO)

**S1 Data.**
(XLSX)

## Author Contributions

**Conceptualization:** Atefeh Ahmadi, Mahtab Mehrabbeik.

**Data curation:** Sourav Roy.

**Formal analysis:** Atefeh Ahmadi, Sourav Roy, Mahtab Mehrabbeik.

**Funding acquisition:** Matjaž Perc.

**Investigation:** Atefeh Ahmadi, Sourav Roy, Mahtab Mehrabbeik.

**Methodology:** Atefeh Ahmadi, Sourav Roy, Mahtab Mehrabbeik.

**Project administration:** Dibakar Ghosh, Sajad Jafari, Matjaž Perc.

**Resources:** Atefeh Ahmadi, Mahtab Mehrabbeik.

**Software:** Atefeh Ahmadi, Sourav Roy, Mahtab Mehrabbeik.

**Supervision:** Dibakar Ghosh, Sajad Jafari, Matjaž Perc.

**Validation:** Sourav Roy, Dibakar Ghosh, Sajad Jafari, Matjaž Perc.

**Visualization:** Dibakar Ghosh, Sajad Jafari, Matjaž Perc.

**Writing – original draft:** Atefeh Ahmadi, Mahtab Mehrabbeik.

**Writing – review & editing:** Dibakar Ghosh, Sajad Jafari, Matjaž Perc.

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
