## [Decision Letter · Decision Letter 0]

2 Jan 2023

PONE-D-22-31119The dynamics of a duopoly Stackelberg game with marginal costs among heterogeneous playersPLOS ONE

Dear Dr. Ghosh,

Thank you for submitting your manuscript to PLOS ONE. After careful consideration, we feel that it has merit but does not fully meet PLOS ONE’s publication criteria as it currently stands. Therefore, we invite you to submit a revised version of the manuscript that addresses the points raised during the review process.

We look forward to receiving your revised manuscript.

Kind regards,

Xiaojie Chen

Academic Editor

PLOS ONE

Journal Requirements:

Reviewers' comments:

Reviewer's Responses to Questions

**Comments to the Author**

1. Is the manuscript technically sound, and do the data support the conclusions?

Reviewer #1: Yes

Reviewer #2: Yes

2. Has the statistical analysis been performed appropriately and rigorously? 

Reviewer #1: N/A

Reviewer #2: N/A

3. Have the authors made all data underlying the findings in their manuscript fully available?

Reviewer #1: Yes

Reviewer #2: Yes

4. Is the manuscript presented in an intelligible fashion and written in standard English?

Reviewer #1: Yes

Reviewer #2: Yes

5. Review Comments to the Author

Reviewer #1: Authors proposes a duopoly Stackelberg game theoretical model with marginal costs where players are heterogenous. The paper includes mathematical analysis and numerical results investigating the dynamics and stability of interactions of the model.

The research problem addressed is interesting, novel and important, and is highly relevant to the audience of PLOS ONE. Stackelberg game is an important framework (in economics and multi-agent systems literature) to model sequential interactions with imperfect information. Authors have extended the model with to capture important features including heterogenous players and marginal costs, in order to better capture real-world interactions. The mathematical analyses in the paper are carried out in a clear and competent manner, supported also by suitable numerical simulations.

There are only a few suggestions that the authors might consider to make further improvements:

In game theoretical literature, heterogeneous factors in the interactions can be modelled through heterogenous networks, see surveys in i”) Statistical physics of human cooperation." Physics Reports 687 (2017): 1-51. ii) “Evolutionary games on graphs." Physics reports 446.4-6 (2007): 97-216. It would be useful to discuss these relevant works, e.g. whether such heterogenous factors can be captured in the current works (with further extensions).

Moreover, there exist recent game theoretical models studying dynamics of competition and coordination among firms producing goods; see e.g. i) "Evolution of coordination in pairwise and multi-player interactions via prior commitments." Adaptive Behavior 30.3 (2022): 257-277; ii) “Shake on it: The role of commitments and the evolution of coordination in networks of technology firms." ALIFE 2022: The 2022 Conference on Artificial Life. MIT Press, 2022. Also, it’s been studied how incentives such as reward and punishment can help achieve safety and ethical behaviours in technology development competition among firms, see e.g. "Mediating artificial intelligence developments through negative and positive incentives." PloS one 16.1 (2021): e0244592.

It would be useful to discuss these relevant works in terms of game theoretical research into firms competition in a single market.

Minors / typos:

- line 16: those —> that

- line 65: in order to propose the

- line 118: remove Systems (13)

- Change Systems (13) to Equation (13), in all places.

Reviewer #2: In this work, the authors study the duopoly Stackelberg model, a very important model in the econ game theory. To make the model more suitable for real-world scenario, the duopoly Stackelberg model is extended to the case of two heterogeneous players, i.e., rational and adaptable. The proposed mathematical framework also unveils several dynamical aspects, starting from finding the equilibrium points to their stability analysis. With the consideration of marginal cost, this work achieves a breakthrough in this topic as this realistic case is still unexplored yet.

1. A variety of experiments, such as one-dimensional and two-dimensional bifurcation diagrams, Lyapunov exponents spectra, and Kaplan-Yorke dimension are leveraged to examine the influence of model parameters, ensuring the completeness of this work. The results support the conclusion of this manuscript.

2. To make the manuscript more comprehensive, the authors can discuss more about the potential future work before wrapping up the study.

3. The manuscript is well written and easy to read. I recommend acceptance in PLOS ONE.

6. PLOS authors have the option to publish the peer review history of their article (what does this mean?). If published, this will include your full peer review and any attached files.

Reviewer #1: No

Reviewer #2: No

---

## [Author Response · Author response to Decision Letter 0]

6 Feb 2023

Review Comments to the Authors

Reviewer #1: 

Authors proposes a duopoly Stackelberg game theoretical model with marginal costs where players are heterogenous. The paper includes mathematical analysis and numerical results investigating the dynamics and stability of interactions of the model.

The research problem addressed is interesting, novel and important, and is highly relevant to the audience of PLOS ONE. Stackelberg game is an important framework (in economics and multi-agent systems literature) to model sequential interactions with imperfect information. Authors have extended the model with to capture important features including heterogenous players and marginal costs, in order to better capture real-world interactions. The mathematical analyses in the paper are carried out in a clear and competent manner, supported also by suitable numerical simulations.

There are only a few suggestions that the authors might consider to make further improvements:

Comment 1: In game theoretical literature, heterogeneous factors in the interactions can be modelled through heterogenous networks, see surveys in i") Statistical physics of human cooperation." Physics Reports 687 (2017): 1-51. ii) "Evolutionary games on graphs." Physics reports 446.4-6 (2007): 97-216. It would be useful to discuss these relevant works, e.g., whether such heterogenous factors can be captured in the current works (with further extensions).

Response 1: Thank you for this comment. The introduction has been enhanced as suggested by the reviewer and with the recommended references. We have also discussed a few additional related works.

Comment 2: Moreover, there exist recent game theoretical models studying dynamics of competition and coordination among firms producing goods; see e.g., i) "Evolution of coordination in pairwise and multi-player interactions via prior commitments." Adaptive Behavior 30.3(2022): 257-277; ii) "Shake on it: The role of commitments and the evolution of coordination in networks of technology firms." ALIFE 2022: The 2022 Conference on Artificial Life. MIT Press, 2022. Also, it's been studied how incentives such as reward and punishment can help achieve safety and ethical behaviours in technology development competition among firms, see e.g., "Mediating artificial intelligence developments through negative and positive incentives." PloS one 16.1 (2021): e0244592.It would be useful to discuss these relevant works in terms of game theoretical research into firms competition in a single market.

Response 2: Thank you for pointing this out. The introduction has been improved considering the reviewer's suggestion and the recommended references.

Comment 3: Minors / typos:

- line 16: those —> that

- line 65: in order to propose the

- line 118: remove Systems (13)

- Change Systems (13) to Equation (13), in all places.

Response 3: The reviewer is correct. Sorry for the English language mistakes. We have polished the manuscript carefully.

 

Reviewer #2: 

In this work, the authors study the duopoly Stackelberg model, a very important model in the econ game theory. To make the model more suitable for real-world scenario, the duopoly Stackelberg model is extended to the case of two heterogeneous players, i.e., rational and adaptable. The proposed mathematical framework also unveils several dynamical aspects, starting from finding the equilibrium points to their stability analysis. With the consideration of marginal cost, this work achieves a breakthrough in this topic as this realistic case is still unexplored yet.

A variety of experiments, such as one-dimensional and two-dimensional bifurcation diagrams, Lyapunov exponents spectra, and Kaplan-Yorke dimension are leveraged to examine the influence of model parameters, ensuring the completeness of this work. The results support the conclusion of this manuscript.

Comment 1: To make the manuscript more comprehensive, the authors can discuss more about the potential future work before wrapping up the study.

Response 1: We agree with the reviewer's assessment. As suggested by the reviewer, the discussion and conclusion section has been enriched by including potential future works.

The manuscript is well written and easy to read. I recommend acceptance in PLOS ONE.

We tried our best to improve the manuscript and made the required changes. We appreciate the warm work of the editors and reviewers and hope that the correction is accepted. Once again, thank you very much for your comments and suggestions.

---

## [Decision Letter · Decision Letter 1]

16 Mar 2023

The dynamics of a duopoly Stackelberg game with marginal costs among heterogeneous players

PONE-D-22-31119R1

Dear Dr. Ghosh,

We’re pleased to inform you that your manuscript has been judged scientifically suitable for publication and will be formally accepted for publication once it meets all outstanding technical requirements.

Kind regards,

Xiaojie Chen

Academic Editor

PLOS ONE

**Comments to the Author**

Reviewer #1: The authors have addressed all my previous comments very well, with love to details. I would be happy to recommend publication of the paper in its present form.

---

## [Editor Report · Acceptance letter]

24 Mar 2023

PONE-D-22-31119R1 

The dynamics of a duopoly Stackelberg game with marginal costs among heterogeneous players 

Dear Dr. Ghosh:

I'm pleased to inform you that your manuscript has been deemed suitable for publication in PLOS ONE. Congratulations! Your manuscript is now with our production department. 

Kind regards, 

on behalf of

Professor Xiaojie Chen 

Academic Editor

PLOS ONE